# Current Update on Transcellular Brain Drug Delivery

**DOI:** 10.3390/pharmaceutics14122719

**Published:** 2022-12-05

**Authors:** Bhakti Pawar, Nupur Vasdev, Tanisha Gupta, Mahi Mhatre, Anand More, Neelima Anup, Rakesh Kumar Tekade

**Affiliations:** Department of Pharmaceutics, National Institute of Pharmaceutical Education and Research (NIPER), Ahmedabad, Opposite Air Force Station Palaj, Gandhinagar 382355, Gujarat, India

**Keywords:** blood–brain barrier, transcytosis, nanocarriers, nose-to-brain delivery, exosome, peptide

## Abstract

It is well known that the presence of a blood–brain barrier (BBB) makes drug delivery to the brain more challenging. There are various mechanistic routes through which therapeutic molecules travel and deliver the drug across the BBB. Among all the routes, the transcellular route is widely explored to deliver therapeutics. Advances in nanotechnology have encouraged scientists to develop novel formulations for brain drug delivery. In this article, we have broadly discussed the BBB as a limitation for brain drug delivery and ways to solve it using novel techniques such as nanomedicine, nose-to-brain drug delivery, and peptide as a drug delivery carrier. In addition, the article will help to understand the different factors governing the permeability of the BBB, as well as various formulation-related factors and the body clearance of the drug delivered into the brain.

## 1. Introduction 

As stated by the American Brain Foundation, one in six persons suffer from brain illnesses, and the annual expense of treating these conditions is over USD 1 trillion. As a result, there is an increasing demand for the discovery of effective therapies for brain diseases. The blood–brain barrier (BBB) plays a critical role in protecting and maintaining optimal conditions in the highly regulated microenvironment of the central nervous system (CNS) [1]. However, the presence of the BBB also limits the entry of most pharmaceutical drugs. Overall, 98% of CNS-acting small molecules and 100% of large molecules, such as proteins, peptides, siRNA, monoclonal antibodies, etc., do not penetrate the brain at an effective concentration [2,3]. Multiple factors govern the permeability of drugs across the BBB. Examples of such factors are mechanical stress, tight junctions (TJ), the release of cytokines, and diseased states such as neuroinflammation, aging, etc. In the next section, we comprehensively discuss these factors and their role in brain drug delivery. 

Drug delivery to the brain often occurs via two major pathways: transcellular (through the cells) and paracellular (between adjacent cells). Ions and solutes travel via the paracellular route using concentration gradients to diffuse passively through the BBB. The transcellular route comprises different mechanistic pathways, such as passive diffusion, carrier-mediated transport, and transcytosis. Transcytosis is further classified into two major pathways: receptor-mediated transcytosis (RMT) and adsorptive-mediated transcytosis (AMT). The transferrin receptor, low-density lipoprotein (LDL) receptor, and insulin receptor are some of the most researched targets for RMT in brain endothelial cells [4]. Recent developments in RMT offer methods to get beyond the BBB and create more effective drug delivery to the brain. In addition to these mechanistic routes, brain drug delivery also takes advantage of different transporters, such as organic anion and cation transporters [5].

Thus, while designing the formulation, one should consider all the parameters associated with the drug, such as particle size, molecular weight, the solubility of the drug, surface charge distribution, etc. [6]. The development of nanotechnology has given rise to various nanocarriers with particle sizes ranging from 1 to 100 nm, among which polymeric nanoparticles (PNPs), solid lipid nanoparticles (SLNPs), liposomes, and micelles were introduced as nanocarriers for the treatment of different neurological disorders as shown in Figure 1. However, more recent and sophisticated nanocarriers, including dendrimers [7], PNPs [8], self-assembled micelles [9], prodrugs [10] and exosomes [11], among others, have demonstrated considerable promise over earlier delivery systems. In this current review, we have discussed the novel nanomedicines used in transcellular brain drug delivery.

## 2. Crossing the BBB—A Major Obstacle in Attaining Transcellular Brain Drug Delivery

The brain is the body’s most complicated and vital organ. Because of this, it is essential to keep the brain safe from anything that could cause infection, wrong activation, inflammation, or even the death of brain cells [12]. The CNS consists of the brain and spinal cord. The blood–cerebrospinal fluid (CSF) barrier and BBB keep harmful substances out of these spaces. These barriers are selectively permeable to certain substances while blocking unwanted substances from entering the brain [12]. Thus, delivering a drug to such an area of the brain needs a system that can cross such barriers without causing any harm to them. Both bio and small molecules can pass the BBB via various mechanisms, but the two most common are transcellular and paracellular transport [13].

Depending on nature, other small molecules can cross the BBB and reach cell parenchyma in three ways. The first is that tiny hydrophilic molecules are the only ones that can use the aqueous channel for paracellular transport. Yet, it is constrained by the endothelial cells’ tight connections, which can only relax momentarily under regulatory control [13]. The term paracellular transport is used to describe the passage of chemicals between two neighboring endothelial cells. To prevent the passage of big molecules, endothelial cells have intercellular connections. Still, molecules that are narrower than the void can travel through it, which demonstrates a momentary relax [1]. Tight junctions in the endothelium of the blood–brain barrier prevent molecules that are greater than 4 nm in size from passing through, whereas smaller molecules and water-soluble solutes can enter cells via paracellular transport [14]. Several studies have attempted to break down the intercellular junctions between cells in order to improve the delivery of molecules to the paracellular space using either internal or external stimuli. This barrier can be breached as a result of cerebrovascular diseases and traumatic brain injuries. One study shows that, upon inducing a focal partial transection of the optic nerve, the mammalian BBB transiently opens in the visual centers [15,16].

To continue, it is known that tiny lipophilic compounds can penetrate brain tissue via transcellular diffusion, a non-saturable process. Nevertheless, transcellular diffusion requires molecules to move through the cytosol, luminal membrane, nucleus, and abluminal membrane before arriving at brain tissue. This is problematic since lipophilic chemicals become trapped inside the cell membrane. In addition, if the drug is a substrate for an efflux transporter, the diffusion could be altered [17]. Thirdly, most leftover tiny molecules are delivered to the brain by a substrate-specific mechanism, carrier mediators, and endogenous pathways. They are facilitated by the concentration gradient of the substrates and are assisted by the appropriate transporters [18].

Many factors can lead to poor delivery of drugs to the brain, including physicochemical characteristics of the drug, its size, shape, lipophilicity, molecular weight, efflux pump over activity, the solubility of the drug, etc. Thus, these kinds of factors need to be taken into consideration while formulating a system for delivering to such kind of sites. Fundamentals of the BBB and biomolecular exchange are shown in Figure 2.

## 3. Parameters Governing BBB Permeability and Access to the Brain

As discussed above, BBB is a major barrier to targeting the brain. Various parameters (as shown in Figure 3) governing the delivery of therapeutics to the brain are discussed in the below section.

### 3.1. Mechanical Stress

Compressive, tensile, and shear stress are mechanical stresses that can be generated internally or applied externally with relative ease. Mechanically activated systems are stimulated by mechanical stress in the body that occurs naturally or is imparted by external devices on the body, both of which have a wide range of magnitude [19,20]. Adult brain health is closely correlated with normal hemodynamics, in which people with cardiac issues also experience brain aging [21]. Neural activity is influenced by hemodynamics, and both processes are combined and spatiotemporally synchronized, particularly in the case of excitatory neuron activity [22].

Occludin and ZO-1, two proteins linked to the tight junction of the BBB, were controlled by blood flow in in vitro studies on bovine brain microvascular endothelial cells (BMECs) [23]. The expression would also increase under higher shear stress, and this process is reliant on the cyclic strain. It has been demonstrated that mechanical stress controls other aspects of transcytosis, including cell behavior. The production of nitric oxide (NO) is influenced by shear stress, regardless of intracellular calcium levels. Caveolae and the Hippo pathway, also known as the mechanosensory pathway, have been shown to have a close reciprocal connection. For YAP/TAZ expression, the Cavin1 and Cav1 transcription factors are regulated by caveolae, suggesting that caveolae are involved in regulating cell mechanosensory activity. By controlling miRNA expression, shear stress can also influence vessel growth [24].

Mechanical stress effects on cell permeability have been thoroughly investigated in some studies. One example showed that fluid shear stress (FSS) alters apical endocytosis of the proximal tubule (PT) through an mTOR-dependent pathway. FSS stimulates PT cell differentiation through two distinct pathways that increase endocytosis as well as ion transport potential. In vivo glomerulotubular homeostasis may be aided by quick optimization of endocytic response by modifications in FSS [25]. Based on the binding of specific epitopes, shear stress modulates Platelet-Endothelial Cell Adhesion Molecule-1 (PECAM1)-mediated endocytosis in HUVECs in several different ways [26].

Music or sound effects on the BBB have been studied by scientists, and they have found that these sound interventions help to open the BBB and enhance functioning of the meningeal lymphatic system. As a result of these interventions, the BBB was opened at a specified frequency limit and volume range measured at 370 Hz and from 90 to 110 dB, respectively. Scientists have evaluated the effect of sound on the changes of blood flow in the cerebrum. The study was performed on the two-month-old mice and they were categorized into four groups, such as control group (without sound), with sound after 90 min, 4 h and 24 h. The opening of BBB was done through the effect of audible sound at frequency 370 Hz and volume of 110 dB in a pulse manner for 2 h The analysis of BBB permeability was performed. The group of sound effect for 90 min showed 23.3 times enhancement in the BBB permeability compared to control group [27]. 

### 3.2. Basement Membrane and Junction Remodeling

Basement membranes are complicated extracellular matrix (ECM) protein layers that separate epithelial and endothelial cells from the tissue beneath. The vascular basement membrane in the CNS plays an. essential role in the development of vessels as well as the formation/maintenance of the blood–brain barrier by separating endothelial cells from neurons/glial cells [28]. Since Claudin-5 is essential for preserving the BBB’s barrier integrity, numerous modalities have been developed to improve drug permeability with this target [29]. Degeneration of occludin is primarily caused by matrix metalloproteinase-2 (MMP-2), while Cav-1 effectively allocates claudin-5 [30]. When siRNAs for claudin-5 and occludin are administered together, the molecules’ permeability can rise by up to 3 to 5 kDa, implying a synergistic effect [31].

Additionally, several initiatives have been produced to target the receptors that are non-junctional and modify BBB TJs. These potential therapeutic modulators are effective, but unlike antibodies or peptides that target TJ proteins, they do not size-selectively loosen the junctions. An example of this modulator is RMP-7, a bradykinin analog that has entered clinical phase 2 for glioma treatment. It did not affect the carboplatin pharmacokinetics at the doses employed in the patient’s trial with recurrent glioblastoma [32,33]. As RMP-7 is an endogenic peptide with low molecular weight, it does not show antigenicity. It only specifically binds to the B_2_ receptor present on the brain capillary endothelial cells, and it primarily enhances the permeability of capillaries in brain tumors. It shows drug selectivity as in the case with carboplatin. RMP-7 has enhanced the levels of carboplatin and loperamide in the brain; however, it did not help chlorethazine or paclitaxel penetrate the BBB [34]. RMP-7 may raise the concentration of these drugs in the brain if they are encapsulated in liposomes, which would easily disguise a few of its physicochemical characteristics during blood transport [35].

There are several ways to keep neurovascular units in direct contact with one another. One such complex is the peg-socket junctional complex, in which the pericytes function as pegs and are incorporated into the sockets of endothelial cells with the help of proteins such as N-cadherin and connexin 43 (CX43) hemichannels [36]. The role of CX43 as a hemichannel among endothelial cells and pericytes has been clarified. It plays an essential role in maintaining intercellular communication among endothelial cells and pericytes, thereby boosting the consistency of barrier properties [37].

Transcytosis is also altered as a result of gap junction modification during tissue remodeling. Through iNOS activity, Cav 3 and CX43 levels in astrocytes will be suppressed throughout the inflammation phase. The gap junction controller CX43 has also been found to interact with Cav1/2, but its relationship with Cav3 still remains unclear [24,38]. Albumin plays an essential role in myeloperoxidase (MPO) transcytosis through caveolae-albumin binding proteins (ABPs) [39]. By binding to adhesion plaques, MPO promotes tissue remodeling and localization to fibronectin along with the induction of ECM nitration [24].

### 3.3. Cytokines

When an injury or infection occurs anywhere within the body, the immune system releases inflammatory peptide substances called cytokines [40]. The BBB is a one-way barrier that allows for the selective transport of various cytokines. Among these are a wide variety of adipokines, as well as the cytokines interleukin (IL)-1α and IL-1β, IL-6, IL-1 receptor antagonist (IL-1ra), leukemia inhibitory factor (LIF), tumor necrosis factor-α (TNF), and ciliary neurotrophic factor. Critical physiological responses to neurodegeneration and inflammation rely on the transported cytokines. The BBB prevents specific cytokines from passing the barrier. Transforming growth factor (TGF)-α tends to accumulate in the vasculature of the brain, whereas TGF-β1 does not penetrate the brain [41].

Intensive research conducted by numerous groups over the last two decades has revealed the direct entry of cytokines via the BBB, which causes significant damage. It is evident that specific cytokines can cross the BBB rapidly and directly within only 30 min of injection. The following mechanisms may facilitate this process: (a) a retrograde axonal transportation system or saturable influx transport (SIT) (IL-1α and α, IL-6, and TNF-α); and (b) circumventricular organs, regions of the brain in which the BBB is inadequate and where cytokines can pass through simple diffusion (GDNF, glial cell-derived neurotrophic factor). Cytokines can also affect the BBB and enhance its penetrability without reaching the brain, for example, by activating and destroying the microvascular endothelial cells’ tight junctions that form the BBB (TNF-α) [41].

Cytokine signaling has been shown to depend on the transcytosis process, and some examples of this phenomenon are described below. IL-11 signaling via IL11R1/2:gp130 complexes occurs on the basolateral and apical sides of polarized cells, whereas IL6 signaling is restricted to the basolateral side via IL-6R:gp130 complexes. Transcytosis, IL-11R1-dependent, transports, and releases basolaterally delivered IL-11 to the apical extracellular environment. As a transcytotic cytokine receptor, IL-11R mediates the transport of IL-6: soluble IL-6R complex and IL-11 across cell membranes [42]. “Suppressor of cytokine signaling 3” (SOCS3), the inducible inhibitor, is crucial for regulating cytokine receptor signaling by efficiently inhibiting JAK-STAT signaling. Cavin-1, found in caveolae, is an essential SOCS3-interacting protein with biological relevance. Loss of cavin-1 increases phosphorylation of cytokine-stimulated STAT3 and eliminates SOCS3-dependent IL-6 signaling inhibition by cAMP, demonstrating the importance of the interaction between cavin-1 and SOCS3 for SOCS3 functioning [43].

### 3.4. Physiochemical Properties

Balanced transcytosis across the BBB has been demonstrated to rely on several physiochemical properties. The pH and protein polarity are essential factors in transcytosis mediated by the transferrin receptor. Endosomal pH modulates iron release by altering the affinity of iron toward transferrin [44]. Multiple intracellular and extracellular pH shifts that result from various mechanisms are caused by neural activity [45]. Modulations in the membrane organization and dynamics of actin, both cholesterol-dependent, are triggered by a change in temperature, suggesting that temperature plays a role in controlling transcytosis [46].

Consistent evidence from the behavior of neurons reveals that endocytosis speed is mainly dependent on temperature, while dynamin and actin mediate temperature-sensitive clathrin-independent endocytosis [47]. CNS medications typically have a lower molecular weight than their non-CNS counterparts, while most CNS drugs have a basic core. H-bonding potential is arguably the most crucial physicochemical property concerning CNS drug design because it influences many properties associated with exposure to the brain. A rise in H-bonding ability has been linked to a decrease in passive permeability, as observed in studies on artificial membranes (PAMPA). Lipophilicity is one of the most crucial physicochemical properties for successful drug discovery and development. Improving in vitro potency by enhancing ligand lipophilicity is a simple and appealing optimization strategy [48].

Although it is possible to optimize the drugs’ physicochemical properties to allow for passive permeation throughout the BBB, chlorambucil tert-butyl ester, after being given in equimolar doses, results in a sevenfold higher proportion in the brain than chlorambucil alone [49]. However, altering the drugs’ physicochemical properties affects their passive diffusion over the BBB and all ADME processes. Several CNS drug discovery initiatives have attempted to increase the delivery of hydrophilic CNS-active molecules by lipidizing a polar parent molecule. This enhanced lipophilicity improved brain drug delivery, but it failed to match in vivo efficacy of such molecules. Discovering drug candidates with the ideal balance of the free portion in plasma and brain is the focus of the new optimization-based method for CNS molecules [50]. There are techniques for curing CNS diseases that involve encapsulating drugs in nanoparticles (NP) to improve their delivery across the BBB without altering the drugs’ physicochemical properties [51].

### 3.5. Age

The downturn in neural function associated with aging is a critical factor in the deterioration of cognitive abilities and the development of many diseases. Aging and adulthood are accompanied by BBB dysfunction, which is associated with inflammatory response and loss of TJs without recruitment of leukocytes. BBB is a strictly regulated interface formed by pericytes, specialized endothelial cells, and astrocytic end feet to shield brain capillaries from harmful substances [52,53].

BBB dysfunction in the aged patient is associated with vascular leakiness and toxic blood-borne substances infiltration into the brain, which may contribute to neural impairments and diseases such as dementia. Clinical evidence on humans has been the mainstay of this theory, showcasing that BBB collapse in the aging population is highly linked with Alzheimer’s disease (AD) and cognitive decline [54]. Traumatic brain injury (TBI) leads to severe breakdown of the blood–brain barrier, causes secondary cognitive impairment symptoms, and raises the likelihood of dementia. DNA damage in the nucleus characterizes cell damage that can directly contribute to aging by enhancing cell dysfunction or indirectly by promoting cellular senescence or apoptosis.

Neurodegeneration is a significant component of age-related pathology and can be defined as the progressive decline of neuronal function and structure, ultimately resulting in neuronal cell death. Many neurodegenerative disorders appear in middle age, and their cognitive and motor symptoms worsen with time and age, possibly shortening patients’ lifespans [55]. Senatorov et al. describe the vascular BBB breakdown in aging rodents and humans, starting in middle age and progressing to old age. Loss-of-function (LOF) and gain-of-function (GOF) experiments demonstrate that the dysfunction of the BBB leads to TGF-β signaling hyperactivation in astrocytes, which is essential and appropriate to create neural dysfunction and pathology related to age in rodents. They also showed that an aged brain may still have a significant latent capacity that can be stimulated by therapeutic TGF signaling inhibition [56].

Another group demonstrates that plasma proteins penetrate healthy brain parenchyma and transcriptional programs specific to the BBB maintain this transport. As people get older, their brains become less efficient at absorbing plasma proteins due to a switch from ligand-specific receptor-mediated transport to non-specific caveolar-mediated transcytosis. Downregulation of the age-upregulated phosphatase ALPL, a probable transport negative regulator, improves uptake of brain transferrin, plasma, and receptor antibodies of transferrin. Transcytosis changes impair physiological transport in the blood–brain barrier with age [57].

### 3.6. Neuroinflammation

Infiltration of circulating leukocytes into the CNS, the generation of inflammatory mediators inside the CNS environment, and CNS-resident cell activation are all examples of different molecular and cellular pathological manifestations that have been collectively referred to as “neuroinflammation”. Neuroinflammation is becoming a common characteristic for targeting many CNS pathologies [58]. BBB permeability induced with neuroinflammation opens a path for circulating pathogens in many brain diseases, including AD, Parkinson’s disease, amyotrophic lateral sclerosis, and multiple sclerosis. When there is inflammation, the BBB is altered, affecting the brain’s immune privilege and introducing neuronal antigens to peripheral inflammatory moieties. These peripheral inflammatory moieties then stimulate the brain’s inflammatory response, hastening the progression of neurological diseases [59].

Targeting inflammation restores the integrity of the BBB, which improves the neuroinflammation associated with neurological disease treatment. Inflammation offers neuroprotective therapeutic potentials against brain diseases. Anti-inflammatory agents have been utilized in treating neurological diseases with an altered BBB. Patients with multiple sclerosis often take glucocorticosteroids, which have anti-inflammatory and immunosuppressive effects, to reduce inflammation and boost BBB function [60]. Restoration of the BBB has also been investigated using therapeutic agents that target inflammatory response and its signaling pathways, including oxidative stress, angiogenesis, inflammatory cytokines, and cytoskeleton reorganization [61].

During neuroinflammation, immune cells infiltrate the brain parenchyma, providing an excellent opportunity for therapeutic agents to cross the BBB. The BBB is thought to limit the number of immune cells entering the brain from the bloodstream. Leukocytes are actively recruited into the brain’s parenchyma when neuroinflammation compromises the BBB [62,63]. This passage throughout the BBB is not a consequence of the BBB’s paracellular permeability, which would permit the passage of relatively small and numerous erythrocytes, causing hemorrhage. Diapedesis is a highly planned and executed process by which immune cells enter the body under normal as well as inflammatory conditions [64,65]. The activation of brain endothelial cells by leukocyte-derived cytokines results in the adhesion molecules expression such as ICAM-1 and VCAM-1, which facilitate leukocyte recruitment. This activation results in leukocyte recruitment. These immune cells can act as a “trojan horse” to deliver therapeutic agents to the brain by crossing the BBB either through chemotaxis or diapedesis [61].

### 3.7. Density of the BBB Receptors

Uchida et al. have studied the proteomics of receptors and transporters present on the BBB [66]. They have performed quantification of the receptors present on the microvessels of the human brain. These microvessels were collected from the brain cortexes of the seven male subjects and with the help of LC-MS, 114 membrane protein expression profiles were studied. The protein expression of a transporter such as breast cancer resistance protein (BCRP) was found to be 8.14 fmol/µg of protein, and 1.85 times the level of expression was found in human versus in mice. Another transporter such as P-glycoprotein was found to be (6.06 fmol/µg) in humans, which was 2.33 times less than mdr1a in mice. Remaining transporters such as organic anion transporter (OAT) and organic cation transporter (OCT) were found to be within limit of quantification (LOQ). Among all these transporters, BCRP was the most extensively expressed protein in humans. This protein is responsible for the efflux of many xenobiotics from the brain toward blood. Temozolamide is a drug majorly used for glioma patients that is effective but that is not transported well by BCRP due to its high efflux efficiency. This can be a reason for the failure of many anticancer agents used for treatment of glioblastoma. Thus, the density of receptors can also influence transcellular brain drug delivery. Another transporter such as low-density lipoprotein receptor-related protein 1 (LRP1), mostly present on microvessels of brain endothelial cells, is also responsible for transcytosis across the BBB. Guo et al. have prepared formulations containing statins-loaded Angiopep-2 derived nanoparticles that help to upregulate the LRP1 transporter, thereby accelerating the process of transcytosis [67].

## 4. Mechanistic Pathways in Transcellular Brain Drug Delivery

The paracellular pathway, which moves drugs intercellularly, and the transcellular pathway, which moves them intracellularly, are the two routes by which chemicals pass across the BBB [68]. Depending on their concentration gradient, solutes and ions pass via the paracellular routes. The paracellular pathways’ tight endothelial cells have structural proteins that make up gap junctions, TJ, and adhesion junctions, allowing for only small water-soluble molecules to pass through them. In contrast, most lipophilic particles, such as steroid hormones and alcohols, can pass transcellularly because they can dissolve through the lipidic membrane. Through endocytosis, giant biological molecules may cross the blood–brain barrier, such as peptides and proteins, while nutrients, amino acids, and glucose are transported across the BBB through receptors, carrier proteins, and vesiculation transcytosis [69,70].

The blood–brain barrier’s tight junctions are crossed by several methods, including carrier-mediated transcytosis, active efflux carriers transcytosis, adsorptive mediated, and RMT [71], as shown in Figure 4. In addition to these methods, it is also possible to boost molecule transport via the BBB by decreasing the efflux of the drugs, decreasing the molecular size, and making them more lipophilic [72].

### 4.1. Receptor-Mediated Transcytosis

Receptors are a group of proteins that include extracellular domains that may connect with particular ligands and transport them inside the cell [73]. The drugs use the endothelial cells’ receptors to act on their target location, specifically bypassing the membranes and limiting their off-target toxic side effects [74].

In the RMT process, the exogenous ligand first binds to its particular receptor. The binding then causes endocytosis, in which the bound ligand and receptor invaginate within the cell to produce an endosome. The generated endosome is subsequently transported across the cell, where the vesicle cargo is exocytosed and delivered to the brain parenchyma [75]. The lipoprotein receptors, transferrin receptors, diphtheria toxin receptors, glutathione transporter, and insulin receptors are the most studied receptors that often participate in receptor-mediated transcytosis to deliver drugs to the brain. RMT is a desirable approach for transporting large heterogeneous molecules up to 80 nm in size from high molecular weight materials of around 80 kDa [76].

The membrane’s luminal or abluminal surface may include the receptors in the RMT system. Luminal side receptors export their ligands from the blood surrounding the CNS into the brain [77]. IgG molecules are transferred from the brain into the circulation in the case of the immunoglobulin Fc receptor, an example of a receptor situated on the abluminal surface [78]. The transferrin receptor, however, is found on both surfaces and causes the transport of its ligand transferrin from the blood into the brain and vice versa [77]. It is typical for the drug-delivering ligand to compete with the endogenous ligand already there. To address this, peptidomimetic monoclonal antibodies are utilized, specific to the receptor, and transport drug molecules without interfering with the endogenous ligand [73].

By serving as a ligand for the transferrin receptor (TfR) found on the cell membrane, transferrin, an 80 kDa monomeric glycoprotein, transfers iron to the cells [70]. In cases of cerebral infarctions, it has been claimed that the transferrin receptor assisted in transporting ginsenoside Rg1 over the BBB [79]. The avidity and affinity of the ligand and receptor interactions must be changed, as well as the stimuli-sensitive connections, to boost the transport of drugs into the brain via the TfR receptor [80]. Since endogenous transferrin already causes the TfR receptor to become saturated, transferrin is not thought to be the best candidate to deliver drugs into the central nervous system (CNS).

To address this, monoclonal antibodies that target the TfR receptor and bind to its extracellular face without interfering with transferrin binding are being investigated [81]. One such murine monoclonal antibody, OX26, has the benefit of binding to several drugs, including BDNF (brain-derived neurotrophic factor), growth hormones, and methotrexate, and it has been proven to be helpful for drug delivery into the CNS. When BDNF was attached to the OX26 antibody using the biotin-streptavidin conjugation method for the treatment of stroke, it was seen that there was a nearly 243 percent increase in motor activity compared to BDNF delivered alone [82]. Another monoclonal antibody, R17-127, is known to react with TfR specifically and is shown to have minimal absorption in the mouse liver and kidney [83]. Using anti-TfR antibodies, a wide range of therapies for different neurological illnesses are being investigated, such as developing chimeric monoclonal antibodies conjugated with TNF to treat the mouse Parkinson’s disease model [84].

The insulin receptor is used to transport insulin over the BBB. However, due to its short half-life and drawback of demonstrating additive hypoglycemia action with endogenous insulin, it is not regarded as a suitable vector to transport medicines into the CNs. When using insulin growth factor or IR monoclonal antibodies, this difficulty does not arise [85]. Human anti-insulin receptor monoclonal antibody was used in an experiment on Rhesus monkeys to shuttle a TNF-α decoy to treat anti-inflammatory diseases [86].

Another receptor is low-density receptor-related protein (LRP1); LRP1 is an excellent receptor to target with drugs for malignant glioblastoma since it is overexpressed broadly in both the blood–brain barrier and in the blood–brain tumor barrier. The ligands that use the LRP1 receptor’s transport mechanism include Apoe, α2-macroglobulin, beta-amyloid, and lactoferrin [87]. Angiopep is a very effective BBB targeting vector carried by LRP1 and is primarily used to deliver drugs into the brain parenchyma to treat glioblastoma [88]. Melanotransferrin, another ligand that uses the LRP1 receptor demonstrated in studies, was readily absorbed by the mouse brain after intravenous (IV) administration and in situ brain perfusion [89].

The diphtheria toxin receptor (DTR) is another highly studied receptor with the extra benefit of being unaffected by endogenous ligands [69]. As a mutant version of the DT with the capacity to bind to the DTR, CRM197 is a ligand that can be employed to transport medicines without being toxic [90]. By boosting the caveolin-1 protein in the cell membrane, which plays a function in the fenestration and invagination of the endothelial cells, CRM197 aids in improving the BBB permeability, and the tight junction proteins can be reformed to enable the drugs to flow through them [91].

Because the creation of the endosome in the second phase of RMT transport may cause the drug to be degraded owing to the presence of lysozymes, the drug can be preserved by utilizing pH-sensitive drug carriers or cationic molecules [92]. Drug distribution through RMT transport depends critically on the affinity of the ligand for the receptor. Low avidity will increase drug release in the brain, but this will require higher drug administration, which is not feasible. High avidity will cause the drug to bind tightly to the receptor and prevent it from traveling to the brain parenchyma [93]. Recently, researchers have designed a formulation to treat AD, preventing the nonspecific targeting of drug and preventing undesired accumulation. In this, nanoparticles are prepared based on dendrigraft poly L-lysines (DGL) siRNA and D peptide, which can target and permeate across the BBB [94]. To attain greater internalization, speedy escape from endosome/lysosomes, and effective transcytosis, this approach links T-peptide, which selectively binds transferrin receptors on the BBB to DGL through acid-breakable long PEG. Tet peptide, which targets damaged neurons particularly, is altered on DGL via shorter PEG. Tet might transport the nanoparticles to the AD after exposure and deliver the drug there. To check the capacity of accumulation into the brain and transcytosis, nanoparticles were labeled with Cy5.5 to visualize fluorescent organ imaging as shown in Figure 5A It was found that T7-attached nanoparticles displayed higher fluorescence as compared to unlabeled T7 nanoparticles. The fluorescence in the brains of the D-DCT, D-DTT, and D-DTCT groups rose throughout the duration, although it was not altered noticeably in the remaining two categories. Compared to the uncleavable group, dual targeting causes more accumulation in the brain. D-DCT7/siRNA and D-TCT7/siRNA are cleavable. Figure 5B shows ex vivo organ imaging in which D-DTCT7/siRNA has greater nanoparticle distribution at 48 h. In other groups, distribution was mainly seen in the liver and kidney. Compared to D-D/siRNA therapy, the brain accumulation of D-DTCT rose to 1.83 times, and that of D-DTT increased to 1.64 times, as shown in Figure 5C. Further, Figure 5D indicates the confocal imaging of the brain with the accumulation of various nanoparticle.

### 4.2. Transcytosis

A lining of astrocytes and tight junctions, created by the tightly packed adhesion of vascularized endothelial cells to one another, make up the blood–brain barrier. Tiny hydrophobic chemicals can diffuse through based on their concentration gradient, whereas small water-soluble molecules migrate via the paracellular route [95]. For molecules moving through a simple diffusion process, a compound’s lipophilicity is crucial. Receptor-mediated transport is essentially for peptides, nutrients, leptin, and insulin. Therapeutics such as azidothymidine and vinca alkaloids pass across the membrane via a carrier-mediated mechanism. Charged species such as cationized serum albumin travel by adsorptive transcytosis. Transcytosis via the BBB is a bidirectional process that involves moving materials from the apical end through the process of endocytosis, then moving them within the cell to the basolateral surface, where they are eventually exocytosed into the brain parenchyma [95]. To be able to move via the transcytosis pathway, a substance must be able to meet a variety of requirements, including being un-ionized, having less than 500 Da as the molecular weight, having a minimum of 10 hydrogen bonds overall, and having a *p* value that is near to 2 [96].

The endocytosis step is a vital part of the transcytosis process, and the clathrin and caveolae-mediated endocytosis pathways are the two major pathways that function at the BBB [97]. In clathrin-mediated transcytosis, the cargo is first endocytosed through clathrin-coated pits and then transformed into cytosolic vesicles due to the GTPase dynamin’s activity, which separates the pit from the plasma membrane. The uncoated vesicle enters the endosomal sorting route following the coat rupture by the ATPase HSC70. This pathway is used by several receptors, including those for insulin and transferrin [98]. On the other hand, the development of caveolae occurs when cytosolic cavins bind to integrated membrane proteins called caveolins [99]. The only major difference between clathrin- and caveole-mediated endocytosis is that clathrins are major proteins in clathrin-coated endocytosis and caveolins are major proteins in caveole-coated endocytosis. The remaining endocytosis occurs in a similar way.

Once the internalized pathways come together to form the early endosomal network, the cargos are sorted intracellularly. The caveolae do not display the clathrin-coated vesicles’ ability to fuse with endosomes. Hence, the caveolae promote the transport of substances such as dextran, albumin, and HRP that do not require receptors. Once the cargo is within the endosomes, it can undergo transcytosis by being transported to the sorting tubules. The lysosomes may break it down as the early endosome develops into the late endosome. The ingested cargo is exocytosed depending on whether the sorting tubules fuse first with the intermediate basolateral sorting endosome or directly with the abluminal membrane [100].

### 4.3. Cell-Mediated Transcytosis

Due to the fact that the body’s cells act as carriers for the drugs, the process is known as cell-mediated transcytosis [101]. Inflammatory conditions in the brain associated with Parkinson’s disease, stroke, AD, and brain tumors cause a variety of lymphocytic cells to move to the site of inflammation through the processes known as chemotaxis and diapedesis [69]. These cells endocytose various colloidal components, such as liposomes and micro- or nanoparticles, and then exocytose the drug cargo to the external medium [102]. As a result, these “trojan horse” cells can ingest a significant amount of micro- or nanocarriers mixed with drug molecules into the brain [69].

Studies have revealed that using monocytes and neutrophils as the carrier systems allowed for the effective delivery of cRGD protein to address the damage caused by cerebral ischemia [103]. RGD-attached magnetic liposomes were administered to the brain of rats in an IL-1β-induced brain inflammation paradigm. The RGD is used as a targeted ligand for the neutrophil- and monocyte-expressed integrin receptors. Using this method, a 16 percent increase in the liposome drug complex was observed. Their uptake in the liver was lessened by this formulation, which provided an additional benefit [104].

The drug must be released from the carrier system at the targeted site, a high dose of the drug should be placed in the carriers, the host cells’ clearance and disintegration should be prevented from affecting the therapeutic substances that are entrapped, there must be effective cell carrier recruitment to the diseased condition, and the entire system must be safe for the patient for the cell-mediated transcytosis approach to deliver the drug successfully [105]. This method differs from others in that it allows for the administration of any molecules, including particulate carrier systems [106]. It has the benefit of extending drug circulation because of the carrier molecules’ inherent capacity to stay in the circulatory system for a prolonged time [107].

Additionally, it reduces cytotoxicity and immunogenicity profiles and directs the medicine directly to the diseased location. Despite these benefits, cell-mediated transcytosis has significant disadvantages, such as premature release from the drug-laden carrier and the occasional inability to reach the site of action with specificity [105].

### 4.4. Adsorptive Mediated Transcytosis

Due to the anions present on the surfaces of the BBB endothelial cells under normal conditions, their surfaces are negatively charged. Because of the membrane’s anionic nature, cationic chemicals can pass through without resistance [108].

Sialoglycolipids, sialoglycoproteins, and the presence of sulfate groups give the luminal side of the cell membrane its distinctive anionic nature. The groups that give the basolateral side its anionic character are chondroitin sulfate proteoglycans and heparan sulfate proteoglycans [109]. Hence, the luminal side’s anionic surface is in charge of moving cargo inside the cell, while the basolateral membrane’s anionic side is used to exocytose the cargo. Since the adsorptive mediated process is based on the direct contact of the drug with the negative charge of the cell’s phospholipid membrane, the physicochemical qualities of the pharmaceuticals are significant to determine the effectiveness of overcoming the biological barrier.

By conjugating different drugs with cationized molecules such as albumin [110], chitosan [111], or a cationized polymer [112], the BBB can be crossed by these drugs. Proteins can be cationized either through genetic fusion or conjugation with a cationic import peptide, or they can be cationized directly by transforming their carboxyl groups into primary amino groups [113]. The cationic penetrating peptides typically range from 10 to 27 amino acids. The positively charged peptides that are typically used in transporting small and large molecules by conjugating them include Poly-arginine (Arg9), TAT peptide, and Syn-B peptide [108]. Once the anionic spots on the membrane have been identified, the proper amino acid sequence for cationic vector design may be constructed. The cationic charge density and the molecular weight of the peptide vectors must also be taken into account to increase the delivery via the AMT process [113].

The first protein to exhibit membrane-penetrating capabilities was the trans-activator of transduction (Tat) protein, a protein from the human immunodeficiency virus [114]. The studies that were conducted in which NR2B96, a peptide molecule, was able to move through the BBB with the assistance of Tat to reduce the infarction volume after stroke, demonstrate Tat’s ability to transport the cargo across the BBB as well as translocate it to its final intracellular target across the neuron plasma membrane [115]. Although more effective than NR2B96, the PSD-95-inhibitor has been shown in recent research to lower infarction volume, and this drug was also able to penetrate the BBB with Tat’s assistance [114]. Another cell-penetrating peptide (CPP) agent, dNP2, was employed to penetrate the BBB with a potent drug called CTLA to treat inflammatory disorders [116].

Another common carrier for delivering medications across the BBB via the AMT pathway is positively charged polysaccharides [117]. Chitosan-drug conjugation results in both a sustained buildup of nanocarriers in brain cells and high penetrability [110]. Additionally, chitosan exhibits significant mucoadhesion, low toxicity, biodegradability, biocompatibility, and paracellular permeability [111]. Cationized albumin can deliver drugs across the BBB by creating nanoparticles [118] or by using them as shell particles [110]. This will increase cell uptake, as albumin serves as a nutrient source for cell proliferation, and because of its cationic nature, it will increase the transendothelial rate.

Although the cationic character facilitates the passage of the cargo through the cell membrane, this property can also be harmful since the positive nature may demonstrate cytotoxicity due to rapid cellular absorption, which will release a large number of cytotoxic chemicals. Reticuloendothelial systems may also catch the carrier cells in the liver and lung if the electrostatic contact is non-specific [117].

### 4.5. Efflux Pumps and Drug Transport

To efflux foreign chemicals out of the brain parenchyma, several efflux pumps are found on the endothelial cells of the brain [119]. Breast cancer resistance protein (BCRP/ABCG2), P-glycoprotein (P-gp/ABCB1), and multidrug resistance protein (MRP1/ABCC1) are the primary efflux pumps that are in charge of expelling undesired compounds. The ATP that is linked to the ABCs provides the energy necessary for the efflux to translocate the substance across the endothelial membrane. The most researched protein, P-gp, is known to inhibit the ability of several medications, including doxorubicin, paclitaxel, vinblastine, valinomycin, and pepstatin, to accumulate in the CNS [120].

Different tactics are used to lessen the interaction between the efflux proteins and the drug molecules since efflux decreases the therapeutic efficacy of the drug molecules. These techniques include altering the drugs chemically to reduce their affinity for the transporter or combining the drug with other compounds that will block the efflux pumps [121]. For instance, because of its capacity to lower efflux activity, the chemically altered version of taxane known as TX-67 was more effective than the original medication paclitaxel [122]. Similarly, the efflux activity is decreased in the analogs of paclitaxel created by adding amine or carboxyl groups at the C7 position [123]. Alternately, to reduce the contact between the pump and the drugs, the drug might be contained in a nanocarrier system or liposomes [124].

The organic anion transporters (OAT1 and OAT3), which are found on the brush boundary of the choroid plexus and which were also observed to be present in an in vitro BBB model, are some additional receptors that are in charge of transporting the molecules [125]. Dicarboxylates, amino hippurate compounds, and β-lactam antibiotics have all been transported by these transporters [126]. Similarly, organic cation transporters (OCT2) are expressed in the brain. However, instead of the BBB, these transporters are known to influence neurotransmitters in neurons [127]. Pyruvic and lactic acids are transported by additional transporters known as monocarboxylic acid transporters (MTC) [128].

## 5. Formulation Consideration in Developing Transcellular Brain Drug Delivery Device

Several factors must be considered when a drug is administered to affect brain sites, as shown in Figure 6. Considering the BBB’s complexity, optimizing the formulation so that it can either bypass the BBB or transport the drug across it is crucial. To achieve this, the basic physiology of the BBB and the physicochemical characteristics and tics of the drug play crucial roles. Particle size, the drug’s lipophilicity, solubility, surface charge, etc., have to be considered thoroughly.

### 5.1. Particle Size

Nanoparticles have received a lot of interest in the recent decade for their potential as drug delivery agents that can transfer drugs throughout the BBB (because of their size-related advantages) and enhance brain medication uptake where it is needed [129]. As a result of nanoparticles, medicine can stay in the bloodstream for longer. This makes it easier for drugs to pass the BBB by interacting with molecules present on the luminal surface of endothelial cells of the brain.

Nanoparticles’ capacity to traverse the BBB can be enhanced by adjusting their size, shape, and type of ligand conjugated [130]. Surface charge and particle size have been shown to play crucial roles in the diffusion of NPs across the brain tissue. It is well known that smaller particle size offers a greater surface area that increases the absorption of the drug at endothelial sites of the brain and enhances the dissolution rate of the drug. As per one study, the average extracellular space (ECS) width in brain tissue is between 38 and 64 nm. In addition, the experimental data demonstrated an inverse relationship between NP particle size and diffusion coefficient. However, these findings only apply to NPs without charges [131]. In another study, coated nanoparticles having a charge of around −5 mV of 114 nm in diameter were able to cross human and rat brains. This shows that particle size, charge, and type of coating can affect the transport of drugs across the endothelial parenchyma [132].

### 5.2. Solubility of the Drug

The solubility of the medicine is a critical component that plays a role in determining the rate and the amount of drug absorption, especially when the drug has to deliver from the nose to the brain route. Since nasal secretions are more watery than other types of secretions, a medicine must have a suitable aqueous solubility to achieve more significant dissolution [133]. This is why most of the time, adequate drug delivery of the drugs and nasal formulations are in molecular dispersion form. Compared with gastrointestinal fluid (GIT), the volume of intranasal secretion is less, which can be a limiting factor in a drug with poor solubility and which demands a high dose of the drug to achieve the same efficacy. When a powdered form is inhaled, it first has to dissolve in the nasal mucosa before it can be absorbed.

For the medicine to be effective, it must dissolve in the nasal cavity. Uptake is proportional to molecule size, with greater uptake occurring for smaller molecules. Therefore, enhancing drug solubility is sometimes required for therapeutically appropriate dosage delivery. Multiple strategies exist to improve the solubility of poorly soluble medicines, such as formulating prodrug, cosolvent addition, and using the complexing agent. For instance, cyclodextrin can effectively increase the solubility of the drug. In addition, the salt form of drugs has better drug profiles [134].

### 5.3. Molecular Weight

In nose-to-brain delivery, although there are several benefits to administering medications by the nose, the drugs’ molecular weight is a significant barrier to their effective absorption [135]. It has been found that the molecular weight of medications has a negative correlation with their ability to be absorbed through the nasal passages. Substances dissolved in water with molecular weights below 300 Da are readily absorbed via the nasal membrane. Compared to intravenous delivery, certain compounds (such as butorphanol) have one hundred percent bioavailability. When the molecular weight is more significant than 300 Da, complications develop. Lipophilic medicines larger than 1000 Da have much lower nasal absorption [135]. Some drug formulations have a molecular weight between 1 and 3.4 kDa administered by the intranasal route with bioavailability of around 10 percent, which was approved by FDA.

Regarding clinically significant effects, medication molecules with a molecular weight greater than 1000 Da require specialized drug delivery methods. Through the use of absorption enhancers, medicinal compounds with molecular weights between 1000 and 6000 Da can attain high bioavailability. These enhancers are surfactants that show weight-dependent absorption.

Absorption enhancers are necessary to boost the bioavailability of peptides and proteins having molecular weights greater than 1 kDa. Compounds with molecular weights smaller than 600 Da soluble in lipids may cross the blood–brain barrier depending on their partition coefficient. However, molecular weight is a limiting factor in the permeability of insoluble compounds in lipids or charge(s). Depending on their molecular weight, polar medicines’ nasal uptake rate and extent are modest. Several studies point to the fact that the physicochemical features of polar medicines with a molecular weight of smaller than 300 Da do not significantly impact the penetration of these pharmaceuticals, as they can pass through aqua pores. This is why the molecular weight of a molecule is critical to take into consideration while delivering a drug to brain sites [136].

### 5.4. Lipophilicity

One of the most critical physicochemical factors that restrict the movement of medicines is their lipophilicity. Whenever there is a rise in the lipophilicity of the compounds, there is also a rise in the permeability of the compounds. Although the nasal mucosa exhibits specific hydrophilic properties, the mucosa of the nose is considered a lipophilic type, and this lipid domain is essential to the proper functioning of the barrier. It is well established that lipophilic drugs are absorbed well compared to hydrophilic drugs (for example, propranolol). According to the findings of one study, the lipophilic prodrug of acyclovir had a higher rate of absorption in rats when it was given to them via the nasal route versus acyclovir itself [137].

Lipophilic chemicals can partition into the lipid bilayer that makes up the cell membrane; thus, they have a greater propensity to quickly penetrate biological membranes through the transcellular route and then move throughout the cell after diffusing into it. In animal models, it is observed that lipophilic moieties such as testosterone, naloxone, and buprenorphine are absorbed completely. Several different compounds have established a link between lipophilicity and the rate at which drugs are absorbed through the nose transcellular route [138]. The lipophilicity of steroids directly affected their ability to be transported. For instance, studies on progesterone and its related compounds showed a direct relationship between the absorption of the drug through the nasal transcellular route and its lipophilicity. In another investigation, the researchers concluded that the amount of quaternary ammonium compounds that were absorbed through the nose increases with increasing the compounds’ molecular weight and degree of lipophilicity [136].

### 5.5. Surface Charge Distribution

Surface charge is another crucial factor that needs to be considered while formulating a brain-delivery system. It has been demonstrated that colloidal nanocarriers have significant potential in directing medications to the appropriate tissue with successful intracellular retention. Because acidic glycoproteins include sialic acid, most endothelial cells and the surface of the BBB have an anionic charge [139]. Therefore, cationic nanomaterials have a great affinity toward endothelial cells. Because of its electrostatic attraction, cationic nanomaterials can spend more time in touch with a blood–brain barrier, resulting in nanomaterials entering the brain via endocytic adsorption-mediated transport [140]. In other words, anionic nanomaterials are repelled by the anionic charge of the BBB, but here, size also comes into the picture. A study was conducted where they concluded that both surface charge and particle size influence drug transport in the brain. In addition to cationic nanoparticles, anionic NLCs (nontoxic) with a particle size less than 200 nm have the potential to be efficiently absorbed by the membrane of the brain [141].

## 6. Strategies for Transcellular Brain Drug Delivery

Currently, several nanotherapeutics are being used in transcellular brain drug delivery owing to their characteristics such as nano-size, ability to modify surface properties, biocompatibility, etc. In the next section, we briefly discuss the different novel carriers for brain drug delivery, such as polymeric nanoparticles, micelles, peptides, exosomes, etc.

### 6.1. Nanomedicines for Transcellular Brain Drug Delivery

In recent years, nanotechnology has gained more attention in delivering neurotherapeutics across the blood–brain barrier (BBB). It is an advanced, promising, and cutting-edge transcellular brain drug delivery approach. Their excellent attributes, such as nano-size, distinct physicochemical properties, and their capacity to use biocompatible nanomaterials for surface modifications, are widely used in biomedical applications. Site-specific targeting across the BBB using nanotechnology has the potential to be tailored to perform specific functions as required [142]. Nanomedicines comprise different polymeric or lipidic carriers, containing a different range of medicines. Nanomedicine possesses a size between 50 and 200 nm, which is too large to diffuse freely through the BBB [143]. To solve this problem, specific targeting moieties, such as ligands that imitate the endogenous moiety, are decorated on the surface of nanoparticles. The following section discusses various nanoformulations/nanocarriers such as polymeric nanoparticles, lipidic nanoparticles, and inorganic nanoparticles used in transcellular brain drug delivery.

#### Types of Nanocarriers

Polymeric nanoparticle

PNPs are colloidal dispersions ranging in size from 10 to 100 nm. Usually, PNPs are fabricated by various biodegradable and biocompatible polymers such as polyesters, poly (lactide) (PLA), poly (D, L-lactide -co-glycolic acid) (PLGA), alginate, chitosan, gelatin, and albumin. The release kinetics and in vivo clearance of PNP can be regulated by managing the cleavage bond of polymers [144]. In the next section, we discuss the mostly explored PNP for transcellular brain drug delivery; dendrimers, self-assembled micelles, and entangled solid polymeric nanoparticles.

Dendrimers are highly branched, monodispersed, and three-dimensional in structure and show a high degree of surface functionality and versatility. In dendrimer-based brain drug delivery, the drug molecules cross the BBB through different transporters, such as receptor-mediated transcytosis, carrier-mediated transcytosis, adsorptive-mediated transcytosis, and cell-mediated transcytosis. Drug molecules may be trapped inside the voids of a dendrimer owing to the open nature of dendrimers. Recently, researchers have studied the utilization of hydroxyl polyamidoamine (PAMAM), which has shown the site-specific targeting of small molecules across the BBB (Mishra et al., 2014). In a further study, the same research group studied the effect of generations (size) of the PAMAM on the build-up of the dendrimer in the brain injury animal model. They have seen that the higher the dendrimer generation (6.7 nm), the more the blood circulation period is. Hence, there is better accumulation in the brain compared to the lower generation (4.3 nm) of dendrimers [145].

Polymeric micelles are self-assembled, nanosized, and amphiphilic. To load hydrophobic drugs into a core of a nanoparticle, hydrophobic blocks are built there, while hydrophilic blocks extend across the nanoparticle’s corona to keep the structure stable in aqueous solutions. Owing to their amphiphilic nature, self-assembled micelles form stealthy and stable nanoparticles. A variety of block polymers have been used in the preparation of micelles, such as polylactic acid (PLA)-polyethylene glycol (PEG) [146], Polyacrylic acid (PAA)-PEG (Zhang et al., 2017a), DGL-PEG (Dendrigraft Ploy-L-lysine)-PEG (Yao et al., 2015), etc. Recently, paclitaxel (PTX)-loaded PLA-PEG micelles were formulated, and their effect on glioma in animal models was observed. These micelles were modified with the t-Lyp ligand. The results show that the modified self-assembled micelles had increased accumulation in the brain and internalization into the glioma cells [147]. Another fascinating study involves the fabrication of nature-inspired wormlike micelles with PEG-grafted poly(2-diisopropyl methacrylate) (PDPA) block copolymers (mPEG-b-PDPA). These micelles were degraded in response to the tumor microenvironment, specifically in the injured brain [148].

Intermolecular interactions between various polymers via different bonding, such as Van der Waals or electrostatic interaction, result in entangled polymer nanoparticles, which solidify into nanoparticles. For instance, poly (D, L-lactide-co-glycolide) (PLGA), a copolymer of lactic acid and glycolic acid, is frequently utilized to deliver different medicinal compounds built into PLGA’s core. Zhou et al. prepared the 65 nm PLGA nanoparticle to traverse the BBB. Further, these PLGA nanoparticles were coated with PEG, leading to enhanced circulation time and prevention from RES scavenging [149].

Lipidic nanoparticles

Lipidic nanoparticles such as liposomes are the most widely used nanocarrier due to their low toxicity and high in vivo stability. The hydrophilic compartment at the center of these tiny spherical vesicles is surrounded by one or more phospholipid bilayers, which gives them morphological similarities to cell membranes. They are an intelligent method for delivering drugs, proteins, and peptides [150]. Different attempts have been made to generate long-circulating and targeted liposomes because unchanged traditional liposomes have limited circulation time in the body since they are swiftly cleared from the systemic circulation by the cells of the reticuloendothelial system (RES) [151].

In recent studies, scientists have designed a dual therapeutic approach by preparing a thermoresponsive ferri-liposome to target glioma. Glioblastoma-specific cell-penetrating peptide (P1NS) and anti-GBM antibody (TN-C) were coupled over the surface of the liposome (DOX@P1NS/TNC-FeLP). Moreover, the liposomes were co-loaded with doxorubicin (DOX) and superparamagnetic iron oxide nanoparticles (SPIONs) to produce temperature-responsive drug release when using an alternating magnetic field (AMF). It was observed that P1NS/TNC-FeLP quickly crossed the BBB and presented a drug release profile that was glioblastoma-specific, thermo-sensitive, and responsive to cellular uptake [152].

Inorganic nanoparticles

Inorganic nanoparticles comprise different noble atoms, such as gold, silver, and palladium. These atoms possess distinctive optical, physical, electrical, and chemical characteristics. Gold nanoparticles and magnetic nanoparticles (SPIONs) are the most widely utilized inorganic nanomedicine for biomedical purposes due to their simple manufacturing, excellent biocompatibility, and surface engineering. Recently, cellular internalization of (brain parenchyma) transferrin-modified gold nanoparticles coupled with PEG linkers was studied. Due to the presence of fewer transferrin receptors, they found that gold nanoparticles with a size of 20 nm have shown lower accumulation in the brain compared to the 45 and 80 nm sized gold nanoparticles [153].

### 6.2. Nose-to-Brain Transcellular Brain Drug Delivery

The drug distribution across the BBB employing various nanomedicines that must disrupt the BBB and then demonstrate their therapeutic function was covered in the prior section. However, modern drug delivery methods use a nose-to-brain technique, which is more sophisticated. This nose-to-brain delivery pathway bypasses the BBB and carries the drug to the various regions of the brain, mainly through trigeminal and olfactory neural pathways. Compared to the oral and parenteral routes, this one has several benefits and can deliver drugs directly to the brain, where they can be used to treat a variety of central nervous system diseases such as psychosis, Parkinson’s, and Alzheimer’s disease [154]. The delivery of different biomolecules via the nose to the brain is shown in Figure 7.

A prospective drug for nasal administration to the targeted brain has been studied: quercetin, a flavonoid with neuroprotective properties. Due to its low water solubility and high first-pass metabolism, quercetin has a low oral bioavailability. Manta et al. prepared the cyclodextrin (methyl-β-cyclodextrin, Me-β-CD, and hydroxypropyl-β-cyclodextrin HB-β-CD) complexation with quercetin achieving a 17-40-fold increase in the solubility of quercetin further formulating into lyophilized powder. This powder was further blended with mannitol/lecithin in different ratios to form microparticles, and it has shown promising results for intranasal administration of quercetin formulation [155,156].

Rivastigmine is another example of a drug that is an anticholinesterase used for the treatment of AD. Gao et al. designed a mucoadhesive microsphere of rivastigmine, but this formulation failed to cross the BBB. The microsphere formulation was further modified with lectin. In vivo evaluation was carried out to check the therapeutic efficiency of the formulation, and results have shown that memory retention was improved with a lectin microsphere compared to a plain microsphere [157].

### 6.3. Peptide as Brain Drug Delivery Carrier

Peptides are small biological molecules containing less than 50 amino acids. They are biologically active and have similar functions to proteins. Proteins and peptides have been discovered and utilized recently to pass through the BBB through several transport methods [158]. With the help of different peptide-mediated transporters, drug molecules cross the BBB. Mostly cell-penetrating peptide-Tat (T7) acts as a delivery vehicle or is sometimes conjugated with the cargo molecule. Recently, Kim et al. successfully performed systemic delivery of microRNA-21 antisense oligonucleotides to the brain using T7-peptide decorated exosomes. Thus, this T7-Exo was injected into the rat intracranially, and results showed that it was effectively delivered into the brain. Moreover, T7-Exo reduced the miR-21 level in glioblastoma [159].

Al-azzawi et al. designed a biodegradable and biocompatible drug delivery system capable of delivering a higher flurbiprofen cargo across the BBB with a concentration that should minimize the side effects associated with current high-dose therapy for the treatment of AD. This work, in particular, demonstrates the superior efficiency of receptor-mediated transcytosis-mediated flurbiprofen delivery over systems that depend on adsorptive-mediated transcytosis [160]. Li et al. identified the peptide sequence that helps to target the brain utilizing in vivo phage. The randomly 7 mer peptide was intravenously given to nude mice, and phages were recovered from the parenchyma of mice brains. Afterward, stepwise separation of the number of phages and a peptide sequence, PepC7, was obtained. Compared to randomly used peptide, Clone7-1, which encrypts PepC7, showed 41 times higher translocation efficiency. In confocal fluorescence imaging, the Cy5.5-labeled PepC7 displayed a greater capacity of brain targeting, as shown in Figure 8 [161].

### 6.4. Exosome as Brain Drug Delivery Carrier

Exosomes are nanoscale extracellular vesicles of a size range of 30–150 nm released by the majority of cells. Exosomes contain various biomolecules such as nucleic acids, lipids, proteins (lysosomal proteins, surface proteins, fusion proteins, and heat shock proteins), and carbohydrates. Exosomes are more stable than polymeric and lipidic nanoparticles due to their endogenous nature, and they work as a better alternative for drug delivery. Exosomes facilitate pathological and physiological processes by releasing their cargo after internalization.

Several separation methods are used to isolate exosomes, including immunological separation, ultracentrifugation-based separation, size exclusion chromatography, ultrafiltration, magnetic separation, dielectrophoretic separation, acoustic fluid separation, polymer-based precipitation method, and deterministic lateral displacement separation. Exosomes are characterized by dynamic light scattering, transmission electron microscopy (TEM), nanoparticle tracking analysis (NTA), fluorescence correlation spectroscopy (FCS), atomic force microscopy (AFM), surface plasmon resonance detection (SPR), nuclear magnetic resonance, colorimetric detection, and enzyme-linked immunosorbent assay [162,163,164].

Four steps are involved in the generation of exosomes, i.e., budding, invagination, formation of multivesicular bodies, and secretion. Various pathways can be involved in the internalization of the exosomes in the recipient cells. Fusion of exosomes within the plasma membrane can be performed directly, and other pathways such as clathrin or caveolin-mediated endocytosis, micropinocytosis, phagocytosis, and lipid draft-mediated endocytosis can be involved in the uptake of exosomes. Drug resistance can be minimized by bypassing the P-gp efflux transporter using exosome-facilitated drug delivery [165]. Exosomes elicit exceptional features such as low immunogenicity, biocompatibility, proficient internalization, easy functionalization for targeted delivery, and higher loading capacity that attracts researcher recognition as a carrier for drug or gene delivery to the brain. Exosomes are also explored for their diagnostic applications [70].

Various methods are used to load the external cargos to the exosomes, such as liposome-based transfection, electroporation, ultrasound-mediated transport, freeze–thaw cycle, incubation with exosomes, membrane permeabilizer, click chemistry and extrusion technique [166].

Macrophage exosomes can enter the brain parenchyma of healthy animals by crossing the BBB without using the assistance of brain-infiltrating immune cells. Improved interaction of exosome and brain endothelium enhanced exosome accumulation in the inflamed tissues of the brain [167]. Morad et al. investigated tumor-derived exosome internalization by breaching the blood–brain barrier. High spatiotemporal resolution microscopy was used to observe the uptake, and they discovered that the transcellular uptake of exosomes involves the endothelial recycling endocytic pathway [168]. Another study showed exosomes’ effectiveness in loading the superparamagnetic iron oxide nanoparticles (SPIONs) as an imaging agent and curcumin as a therapeutic agent. This imaging and therapeutic agent-loaded exosomes were functionalized with neuropilin-1-targeted peptide to target the exosomes to glioma cells. These exosomes crossed the BBB quickly, and an improved therapeutic and imaging effect was observed with fewer side effects [169].

The efficiency of the exosome to deliver anticancer drugs in the brain of the zebrafish model was also evaluated and showed a reduction in tumor growth markers and fluorescence of xenotransplanted cancerous cells [170]. MiR-210 was delivered to the ischemic brain of a transient middle cerebral artery occlusion (MCAO) animal model using the c(RGDyK) peptide-ligated targeted exosome system. After intravenous administration, this system delivers miR-210 to the ischemic brain’s affected region and offers an angiogenic agent for stroke treatment [171]. Additionally, curcumin was also loaded in the exosome, and intravenous administration of c(RGDyK)-targeted curcumin-loaded exosomes helps reduce cellular apoptosis and inflammatory responses in the lesion region of the ischemic brain [172].

### 6.5. Drug Delivery via Active Transporters in BBB

There are three primary approaches used in the transporter-facilitated drug delivery method: (i) altering the drug’s structure so that a particular BBB transporter will identify it as a substrate, e.g., “pseudo nutrient” and will help to transport the drug across the cell membrane; (ii) conjugating a moiety with the parent drug portraying a nutrient-substrate of the specific BBB transporter, also referred to as a prodrug approach; and (iii) decorating the drug-encapsulated nanomedicine with a moiety depicting the substrate of the specific BBB transporter. [173].

Therapeutics transport through brain endothelial cells is regulated by a variety of influx and efflux transporters expressed at the basolateral and apical membranes. Various proteins from the adenosine triphosphate (ATP)-binding cassette (ABC) and solute carrier (SLC) superfamilies engage in drug transport pathways across the BBB. Generally, ABC transporters play a role in drug removal efflux from the brain to the circulation, and SLC transporters aid in drug influx into the central nervous system [174,175].

#### 6.5.1. Influx Transporters

Over 300 SLC transporter-expressing genes, grouped into 52 families, make up the largest identified membrane transporters superfamily. The substrates of SLC are translocated into the endothelial cells via the luminal membrane and come across the abluminal membrane to avoid sequestration inside the capillary endothelium [176].

The SLC superfamily includes several different transporters that play a role in the uptake and excretion of not just endogenous ions/molecules but also various exogenous drugs. There is a class of transporters known as large neutral amino acid transporters (LATs) responsible for transporting neutral amino acids with large lipophilic substituents on the α-carbon such as triiodothyronine l-tyrosine transport through the biological membranes. LAT is an antiporter for amino acids that uses a 1:1 stoichiometry to import a substrate and export an endogenous substrate from the cell [177].

The SLC superfamily includes a subfamily of genes known as the solute carrier organic anion (SLCO) genes that encode a type of influx transporter known as organic anion transporting polypeptide transporters (OATPs). In humans, there are 11 different OATP members. In other tissues, such as the liver and kidney, this transporter has been revealed to play a crucial role in the uptake of organic anions. Among the several OATPs, OATP1A2 (OATP-A, SLCO1A2) was the first to be associated with endothelial cell expression in the human brain [178,179,180].

The SLC superfamily consists of another subfamily, SLC22, having a class of transporters known as organic cation transporters (OCTs). In the SLC22A family, OCT1 (SLC22A1), OCT2 (SLC22A2), and OCT3 (SLC22A3) are the first functionally described subgroup of transporters. OCTs are named because they primarily identify positively charged weakly alkaline molecules or organic cations [181,182,183].

The third and biggest subfamily of membrane transporters in the SLC22A superfamily is the organic anion transporters (OATs), i.e., OAT1-3, OAT4, OAT7, OAT10, and URAT1 proteins (SLC22A6-9, SLC22A11-13). They play a crucial role in the first phase of renal excretion by mediating the organic anions transport in both directions [184,185]. There are at least eight proton-coupled transporters in the subfamily known as monocarboxylic acid transporters (MCTs; SLC16A). Drug substrates such as atorvastatin, salicylate, g-hydroxybutyrate, nateglinide, and nicotinic acid and endogenous substrates such as lactate transport through the MCT1 [186,187,188]. One of the most prominent secondary superfamilies of transporters, i.e., the major facilitator superfamily (MFS), includes GLUT1 (SLC2A1) as a member of the sugar porter subfamily. GLUTs are the highly prevalent sodium-independent bidirectional transporter facilitating glucose transport through the BBB. Abundantly found glucose transporters have drawn much interest as a possible drug delivery pathway to cross the BBB [189,190,191].

#### 6.5.2. Efflux Transporters

The ABC transporter family comprises primary efflux transporters. Thus far, 48 genes have been identified as belonging to the ABC transporter family (excluding pseudogenes). A–G letters denote the seven subfamilies of the ABC superfamily, and each gene in the ABC superfamily is denoted by a number that comes after the family root and the subfamily letter, such as ABCB1. The two ATP binding cassettes, known as nucleotide-binding domains (NBDs), is the defining feature of all ABC transporters. After NBDs binding with ATP, the energy released during ATP hydrolysis is used to propel the drug transportation across an electrochemical gradient. ABC transporters play a crucial role in cellular detoxification systems due to their unidirectional, dynamic export of drugs and endogenous compounds, including lipids and metabolic products [192,193,194].

The first efflux transporter was P-gp (ABCB1, MDR1). It regulates the entry of its substrate medications across the BBB as an ATP-dependent multidrug efflux transporter. P-gp is a drug efflux pump that is mainly found on the luminal side of endothelial cells of the brain. P-gp can transport various medicines from diverse chemical classes because of its low substrate specificity. Comparatively, blocking P-gp or restriction of the drug through BBB is beneficial to decrease the medication’s side effects or to reposition the medication to other organs [195,196,197].

Drug efflux transporter BCRP (ABCG2) belongs to the ABCG family. BCRP was isolated from a breast cancer cell line resistant to multiple chemotherapeutic agents and was found to expel these xenobiotics. BCRP is considered a facilitator of multidrug resistance in cancer with P-gp due to their overlapping and complementary substrate profiles. Various drugs and endogenous substances are the substrates for BCRP [198,199]. MRP4 (ABCC4) is abundantly expressed at the BBB and is a member of the multidrug resistance-related protein (MRP; ABCC) subfamily. The ABCC subfamily of membrane transporters consists of 12 genes labeled ABCC1-ABCC12. The encoded proteins can be further categorized into three groups: the sulfonylurea receptors (SURs), the multidrug resistance-associated proteins (MRPs), and the cystic fibrosis transmembrane conductance regulator (CFTR) [200,201].

### 6.6. Permeability Enhancers for Brain

BBB restriction for therapeutics can be overcome by developing a new idea to create a temporary opening or breaking the tight junctions to disrupt the BBB integrity. Pharmaceutical delivery can be promoted through BBB by using various permeability enhancers. Hyperthermia can also show effective results for therapeutic delivery by changing the functionality of the cellular structure of tissues using increased temperature [202].

Acoustic energy can be concentrated at a focal point using focused ultrasound, leading to enhanced permeability or disruption of the BBB. Microbubbles are commercialized as a contrast agent for forced ultrasound methods to restrain the effects on the effective area and to reduce the harm to surrounding tissues. These microbubbles interplay with the low-intensity ultrasound, resulting in improved permeability and disruption of tight junctions of the BBB [203,204].

Hyperthermia induced by microwaves or radiofrequency-produced electromagnetic waves is also helpful in enhancing BBB permeability. Due to the extracellular matrix’s excellent conductivity and permittivity in tumor tissue, electro-hyperthermia is a sophisticated hyperthermia method that is thought to be selective [205,206]. Kiessling et al. examined the impact of neodymium: yttrium–aluminum–garnet (Nd:YAG) laser irradiation (lambda = 1060 nm) focally on the cerebral protein synthesis, regional cerebral blood flow, and the permeability of the BBB. They irradiated the 44 rat brain surfaces using a focused laser beam and observed that the laser pulse encouraged BBB disruption locally. Laser-induced BBB disruption increased the permeation of molecules to the brain parenchyma by persuading membranous defects in the endothelium of capillaries [206,207]. In one study, scientists designed a novel technique to in vivo rapidly observe BBB opening brought on by 5-aminolevulinic acid (5-ALA)-mediated photodynamic therapy (PDT), where Evans blue dye served as a marker of BBB permeability. This technique combined spectral imaging with an optical clearing skull window. This optical clearing skull window-based spectral imaging technique provides an inexpensive and easily handled technology for in vivo evaluation of BBB opening progress. The visualization of drug transport to the CNS has tremendous potential. As a result, it is vital for treating brain diseases [208].

One of the most promising chemical strategies for the passive transport of therapeutics into cells is CPPs, also known as membrane translocation sequences or protein transduction domains. CPPs are sequences of peptides that are either cationic or amphipathic, allowing them to cross plasma membranes and enter the brain through the BBB. They have therapeutic potential as molecular delivery carriers for a wide range of therapeutics in brain tissue in the context of central nervous system disorders, including glioma. Various types of CPP used, therapeutics involved, type of cell, and concentration of CPP will decide the pathway involved in the uptake of therapeutics, i.e., endocytic or non-endocytic pathways. When transporting molecules, CPPs have excellent penetration. They are very effective, can adhere to the cell surface, and cause minimal damage to the host cell. Because of their ability to be localized at the subcellular site, CPPs are well-suited for the in vitro and in vivo administration of physiologically active compounds, particularly compounds with poor permeability and high interest, such as nucleic acids, proteins, tiny molecules, and nanoparticles [209,210,211].

With the advancement in techniques, nanomedicines have become a fascinating tool for enhancing the permeability of the BBB using devices or carriers with sizes in the nanometer range. Nanomedicine helps to conquer the limitation of the poorly permeable drug to cross the BBB. The chemical structure of the therapeutics will not hinder the permeation when it is encapsulated in nanomedicines. Biomimetic and physicochemical properties of the nanomedicine guided the permeation of impermeable therapeutics to the BBB [212].

The effect of the laboratory-attenuated rabies virus (RABV) on BBB permeability was evaluated. This study proved that the lab-attenuated RABV increased the permeability of the BBB, but the wild type did not affect the permeability. Inflammatory cytokines/chemokines were enhanced in the RABV-infected mice brains. BBB permeability was increased by RABV infection due to the downregulation of TJ proteins (claudin-5, occludin, and zonula occludens-1) induced by cytokines/chemokines [213].

Recently researcher have found a promising target for transcellular transport across the BBB. A lipidic symporter Mfsd2a is majorly expressed by the microvessels of the BBB, which is a part of the MFS, a major facilitator superfamily [214]. This transporter was found to be a potent inhibitor of transcytosis. It inhibits caveolae-assisted transcytosis, primarily regulating the lipidic profile of endothelial cells of the blood–brain barrier. Thus, targeting Mfsd2a could improve the BBB penetration efficiency. Mfsd2a inhibition significantly accelerated transcytosis across the BBB [215]. In order to enhance movement through the BBB, temporary suppression of Mfsd2a could momentarily speed up RMT. Ju et al. studied the effect of Tunicamycin as an inhibitor of Mfsd2a, and they found enhanced penetration efficiency of their prepared nanoparticle formulation across the BBB via transcytosis [216].

### 6.7. Prodrug Approach for Transcellular Brain Drug Delivery

Prodrug design aims to circumvent the various pharmaceutical, physicochemical, pharmacokinetics, and biopharmaceutical barriers to clinical application presented by the parent drug and enhances the penetration of the therapeutics across biological membranes. The prodrug approach promotes the passive diffusion of the hydrophilic drug by increasing its lipophilicity by masking the polar groups of the hydrophilic drug using amide, ester, or acetylated groups of the prodrugs. The prodrug was anticipated to be converted to the drug by the chemical or enzymatic reaction after reaching the CNS to show its efficacy. The basic approach for site-specific drug delivery via prodrugs includes the following three points: (1) the prodrug must reach targeted tissues in the CNS; (2) site-selective conversion to the prodrugs should be present; and (3) the parent drug should show prolonged retention within the target tissue [217].

Heroin is a prototypical morphine prodrug derived from morphine in which the derivatization of OH groups with acetyl esters occurs. Since it has been derivatized, the lipophilicity of heroin is much more than morphine. Due to increased lipophilicity, heroin became 100 times more potent than morphine. Altering the physicochemical properties of the peptides by prodrugs can help in the permeabilization of the peptides through the BBB. Transport of cRGD peptidomimetics and opioid peptides across biological membranes can be enhanced by formulating the cyclic prodrugs of those peptidomimetics or peptides. Esterases converted those prodrugs to the parent drug after reaching their target site [218]. A selective prodrug for estrogen therapy was developed to target estrogen in the CNS. 10β,17β-Dihydroxyestra-1,4-dien-3-one (DHED) is the prototypical example of this new class of prodrugs for neuroprotection and ocular physiology [219]. The mechanism of prodrug action is shown in Figure 9.

### 6.8. Redox-Responsive Brain Drug Delivery

It has currently been shown that redox-responsive regulation is a crucial intracellular mechanism that influences cell viability, growth, and transformation. Numerous neurodegenerative diseases, including PD and AD, have been linked to redox balance. Recent studies have suggested that oxidative stress is a critical factor in the neuronal loss that leads toward the beginning as well as the development of PD and AD [220]. Li et al. has designed the redox responsive delivery system in which they prepared the mesoporous silica nanoparticles (MSNs) that were responsive to glutathione presented intracellularly. This acted as a drug delivery vehicle, and in the same design, RGD-consisting peptide was added as a porter by forming a disulfide bond.

Moreover, into the mesoporous structure, Doxorubicin was added. This system was found as an effective drug carrier. The peptide layer on the surface of the peptide is separated by the action of intracellular GSH by breaking down the disulfide bond, further releasing the entrapped drug [221] as shown in Figure 10.

### 6.9. Stem Cell as a Brain Delivery System

In targeted drug delivery, such as the injection of RNA to alleviate brain tumors and neurological disorders, stem cells play an essential role [222]. In one study, researchers used MSCs to treat gliomas, and they have also been shown to prevent the growth of glioma cells from enhancing the survival of animals [223]. Animal studies in great numbers suggested that MSC-derived therapy held the potential for cellular treatment. MSCs have also been used to treat other neurological disorders, such as multiple sclerosis and Parkinson’s disease (PD). In the case of multiple sclerosis, an exciting result was seen with the MScs. They have shown remyelination as well as the reduction in the degradation of myelin that existed originally [224]. Parkinson’s disease is a neurodegenerative disease in which dopaminergic neurons present in the substantia nigra are lost. Scientists have studied the effect of MSCs on PD treatment. An encouraging result was that MSCs could have neuroprotective action on dopaminergic neurons due to their anti-inflammatory action, ultimately leading to the speedy recovery of BBB integrity [225].

## 7. Body Clearance of Drug following Brain Drug Delivery

The glymphatic hypothesis has been proposed to explain the extravascular movement of water-soluble compounds that cannot cross the blood–brain barrier. It implies that the fluid in the parenchyma circulating through the interstitium enters via the periarterial route and subsequently leaves by the perivenous route. The term “glymphatic” was created because of the presence of glial end feet that form the boundary of the circulatory system; its flow is thus dependent on the characteristics of these glial cells, while the functions relate to the lymphatics that occur in the body’s peripheral tissues [226]. The perivascular space (PVS), CSF, glial cells, interstitial fluid (ISF), cerebral vascular, and astrocyte aquaporin 4 (AQP4), which regulates the water channels, are the components that play a significant role in the clearance of the medications supplied to the brain [227]. The vascular end feet of the astrocytes generate the low-resistance gap known as the perivascular space. The choroid plexus produces CSF, which is propelled into the perivascular space by the pulsing arteries. The subarachnoid compartment, which is connected to the lateral, third, and fourth ventricles, is where CSF enters the perivascular space [228,229]. The astrocyte aquaporin 4, which is strongly expressed on the astrocytes’ feet, is what permits the CSF to enter the parenchyma of the brain to exchange numerous metabolites and drugs [230]. Finally, through the meningeal lymphatic vessels, the tiny hydrophilic particles and undesirable compounds, together with the CSF fluid, drain into the blood circulation (MLVs).

Although periarterial routes appear to be involved in both the influx and efflux of the solutes, they do not supply the sufficient inflow needed for the solute to be effluxed out. Additionally, it has not been demonstrated that the perivenous pathway controls the outflow. According to the hypothesis, fluid movement causes solutes to be swept out of the parenchyma, but the velocity is too low for the solute particles to successfully diffuse. The classical hypothesis, which was previously stated, made no mention of solute influx through periarterial routes; rather, it proposed that fluid enters the parenchymal space through the BBB, solutes diffuse into the parenchyma, and solutes efflux partially through diffusion and partially through subependymal spaces, perivascular spaces, and white matter tracts. The data are ambiguous as to how the solutes exit the brain and how much of it travels via the parenchyma to the lymph nodes before entering the circulation. Therefore, in order to consider convectional processes other than net flow, both the classical and the glymphatic hypotheses need to be modified [226].

The relationship between drug clearance from the choroid plexus or blood–brain barrier depends on the membrane surface area (S), blood flow function (F), and permeability coefficient (P) of the transport of the drug. All of these variables are governed by the Kety-Renkin–Crone equation, CL = F* (1 − ePS/F) [231]. Typically, the volume of distribution and the efflux clearance value of the drug dictate its half-life in the brain. The drug’s plasma half-life determines the half-life present in the brain if plasma clearance is slower than brain parenchyma elimination. The drug’s half-life in the brain will be greater if plasma clearance is quicker than it is in the brain. Since the brain is a very small component of the body, the clearance of the drug from the brain will only have a minimal impact on the pharmacokinetics of the drug in the blood. However, if administration time of the drug increases, the rate at which it is cleared from the brain parenchyma will have a greater impact on the drug’s half-life in the blood [232].

The BBB also comprises membrane-associated soluble proteases, a prime factor that severely limits the delivery of therapeutics. It is also responsible for catalyzing therapeutics’ metabolism before they actually cross the brain. This factor must also be considered while designing a drug discovery program for the brain delivery of therapeutics [233,234].

## 8. Conclusions

As a result of changes in lifestyle and the deteriorating environment, central nervous system diseases such as AD, Parkinson’s disease, stroke, brain tumors, and neuroinflammation are becoming distressing and devastating to humankind. This review has discussed current methods for drug delivery to the brain. A thorough understanding of the BBB disruption is required to create efficient drug delivery methods for brain diseases. The most recent development in nanotechnology offers a remedy in the form of nanopharmaceuticals, drugs with nanocarriers that can target the brain and cross the BBB. Nanoformulations such as liposomes, polymeric micelles, and dendrimers are most widely used to deliver the drug across the BBB. However, nanomedicines limit their use because long-term use can exhibit toxicity. Recent advancements in brain drug delivery use novel techniques such as nose-to-brain delivery that bypasses the BBB and traverses via the trigeminal and olfactory nerve pathways. More study is required to comprehend and mediate the BBB-crossing mechanisms to enhance the effectiveness of brain delivery techniques using nanotechnology.

## Figures and Tables

**Figure 1 pharmaceutics-14-02719-f001:**
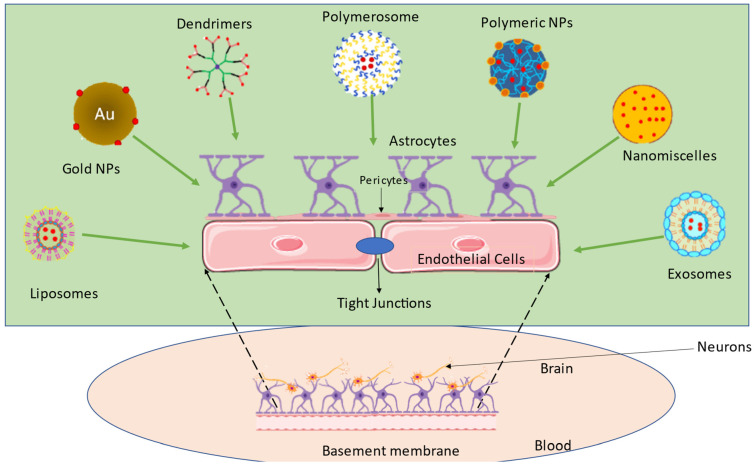
Overview of the blood–brain barrier and current nanotherapeutics for transcellular brain drug delivery.

**Figure 2 pharmaceutics-14-02719-f002:**
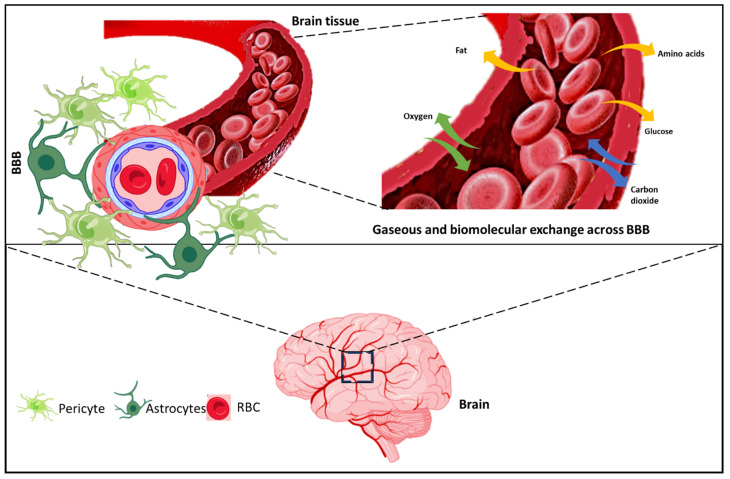
Overview of blood–brain barrier and biomolecular exchange.

**Figure 3 pharmaceutics-14-02719-f003:**
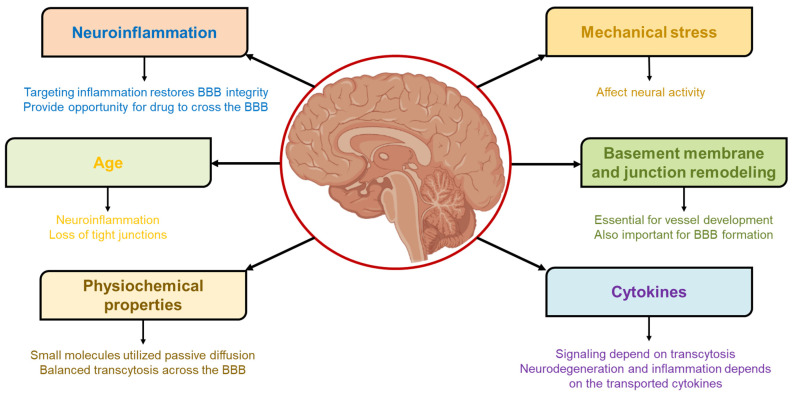
Various parameters influencing brain permeability.

**Figure 4 pharmaceutics-14-02719-f004:**
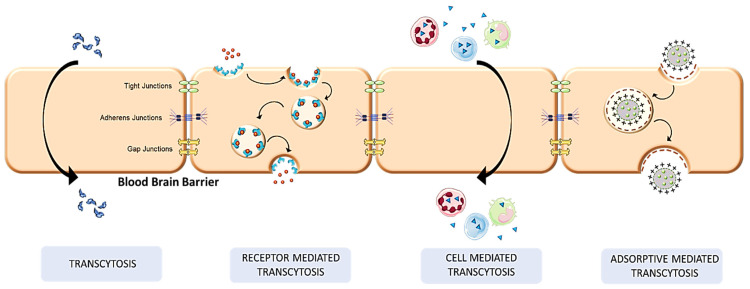
Different mechanistic pathways in transcellular brain drug delivery.

**Figure 5 pharmaceutics-14-02719-f005:**
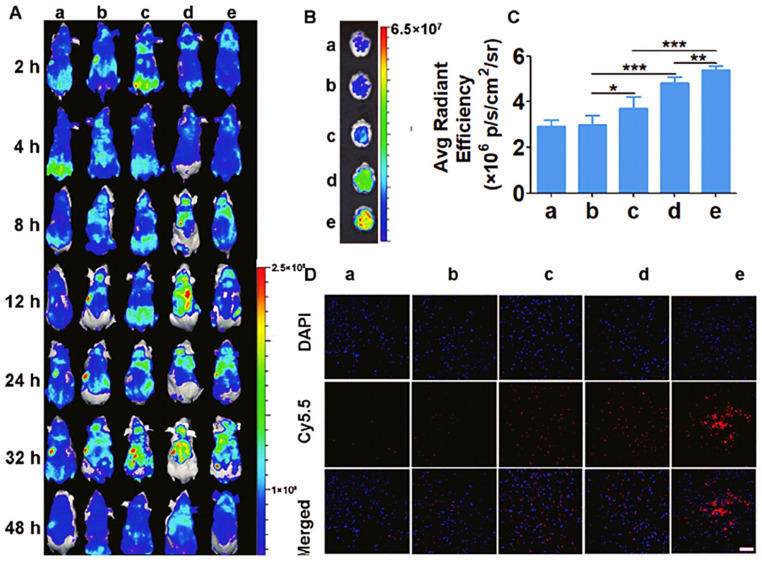
In vivo organ imaging studies. (**A**) Optical imaging of various nanoparticles administered at different time intervals. (**B**) Images of the brain after 24 h of administration. (**C**) Measurement of fluorescence intensity of the brains among different groups at 24 h. (**D**) CLSM images showing the build-up of nanoparticles in the brain. Here: * *p* ≤ 0.05; ** *p* ≤ 0.01; *** *p* ≤ 0.001. (a) D-D/siRNA; (b) D-DT7/siRNA; (c) D-DCT7/siRNA; (d) D-DTT7/siRNA; (e) D-DTCT7/siRNA. Scale bar represents 50 µm. Adopted with permission from [94].

**Figure 6 pharmaceutics-14-02719-f006:**
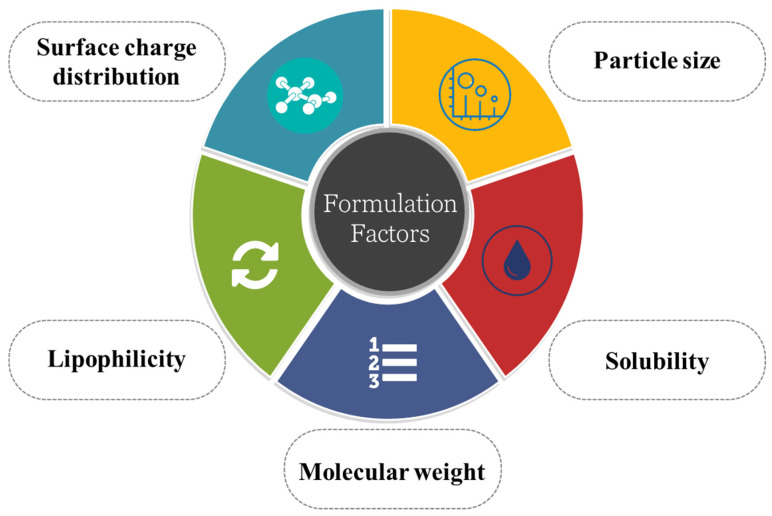
Various formulation factors must be considered when developing the brain drug delivery system.

**Figure 7 pharmaceutics-14-02719-f007:**
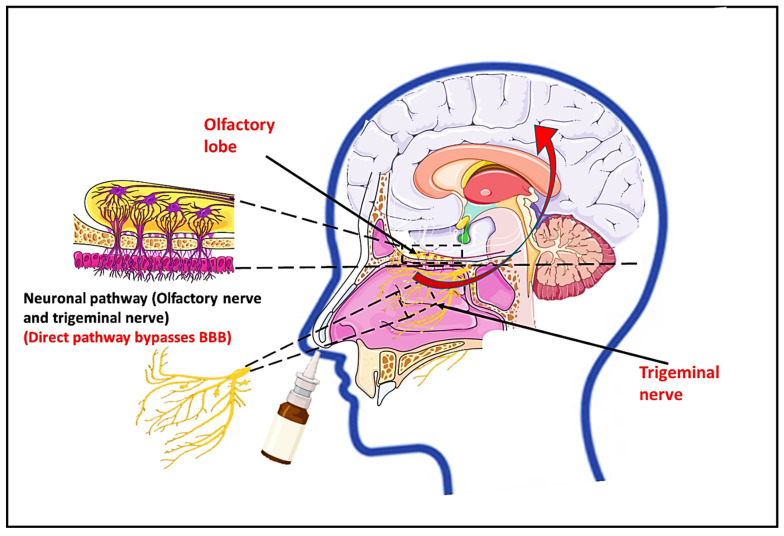
Nose-to-brain drug delivery.

**Figure 8 pharmaceutics-14-02719-f008:**
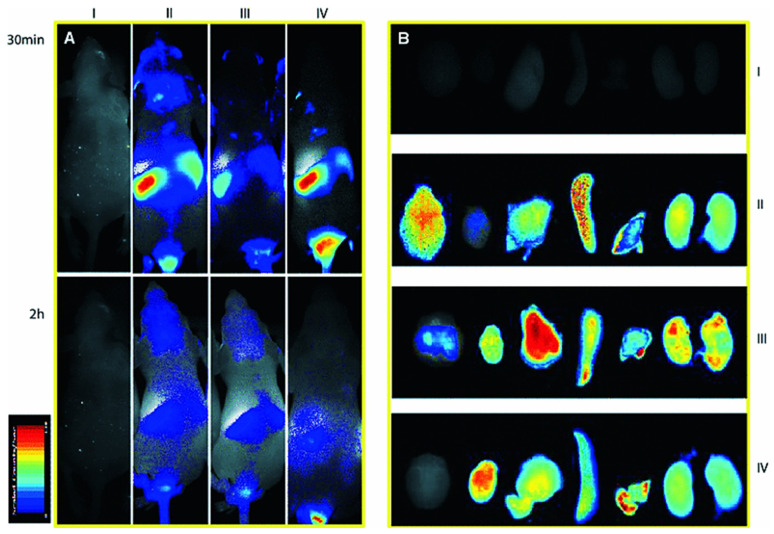
PepC7 in vivo distribution. (**A**) I-Blank control: PBS administered mouse, II-Cy5.5PepC7, III-Cy5.5Pep7, IV-Cy5.5PepSC7. Images were captured after 30 min and 2 h. (**B**) Similarly, ex vivo organ distribution was studied in the same group as listed in A. Adopted with permission [161].

**Figure 9 pharmaceutics-14-02719-f009:**
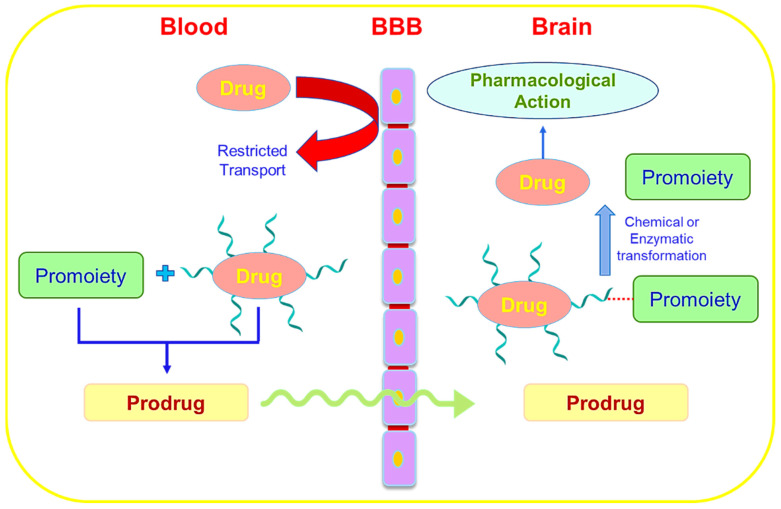
Mechanistic depiction of the prodrug in brain drug delivery.

**Figure 10 pharmaceutics-14-02719-f010:**
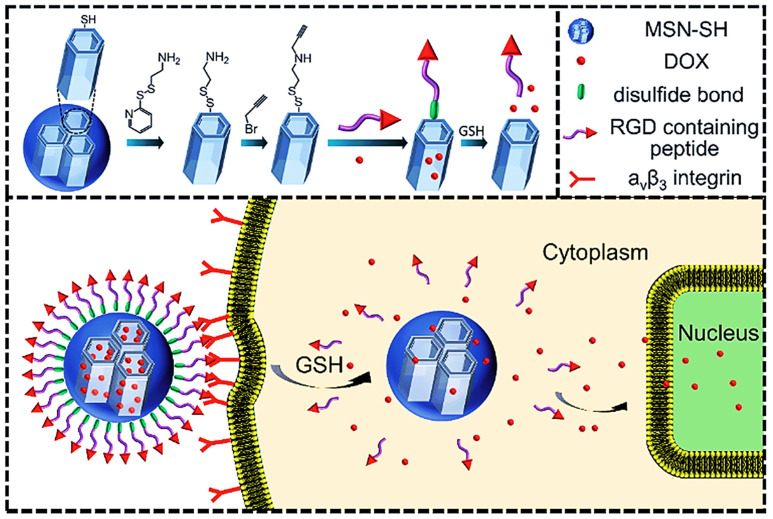
Development of the redox-responsive DOX@MSN-S-S-RGD system. Adopted from [221] with permission under license (CC BY 4.0).

## Data Availability

Not applicable.

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
