# Peer review of "Current Update on Transcellular Brain Drug Delivery"

_pharmaceutics, 2022, doi:10.3390/pharmaceutics14122719_

Round 1

Reviewer 1 Report

In this manuscript, the authors discussed various novel techniques to overcome the blood-brain barrier. In addition, the factors regulating BBB, formulations and clearance are also reviewed. Although the topic is important in brain-targeted drug delivery, there has been many related reviews, which needs this manuscript to be more focused and exhibit some new ideas. Following concerns remain to be addressed before further consideration for publication.

1. The title is “Current update on transcellular brain drug delivery”. So the section of “3. Parameters governing BBB permeability and access to the brain” should be emphasized to the factors on transcellular permeability rather than on both transcellular permeability and paracellular permeability.

2. Membrane protein Mfsd2a regulates the BBB transcytosis rates (Nature 2021 Aug; 596(7872):444-448). There has been reported that targeting Mfsd2a could enhance BBB penetration efficiency (Adv Healthc Mater. 2021 May; 10(9):e2001997). These Mfsd2a-related contents should be added as they are extremely important in transcellular brain drug delivery.

3. The density of BBB receptors also influence the transcellular brain drug delivery efficiency (J Neurochem 2011 Apr; 117(2):333-45; J Control Release 2019 Jun 10; 303:117-129.). These information should be provided.

4. Some related reviews have been published (Pharmaceutics 2021 Nov; 13(12):2024; Acta Pharm Sin B 2021 Aug; 11(8):2306-2325). These reviews should be cited. And the differences should be discussed.

5. Page 8. How gap junctions affect transcytosis is not clearly written. CX43 communicates with Cav-1/Cav-2 in transcytosis. So iNOS can slow down transcytosis by downregulating CX43 in inflammation? The authors should reassemble the language.

6. Page 4. Tight connections between endothelial cells can relax momentarily under regulatory control. The authors should describe this momentary relax more detailed.

7. Page 4. Is there any papers that can overcome the transcellular diffusion in the hydrophilic cytosol after penetrating the hydrophobic membranes?

8. Page 7. Why RMP-7 (which also enhance paracellular BBB permeability) doesn’t size-selectively loosen the junctions? The author should give answers after this argument.

9. Page 12. With gap junctions, TJ and adhesion junctions, which kinds of small water-soluble molecules can pass them? The author should clearly write out the standard. For example, water can pass? Ethanol cannot pass?

10. Page 18. The endocytosis steps, intracellular processes and exocytosis steps are confusing for clathrin and caveolae-mediated pathways.

11. Other very recent brain-targeted drug delivery papers should be cited. Front Mol Neurosci 2022, 15, 895429; Adv Sci 2022, 9(16), e2105854; Fluids Barriers CNS 2022, 19(1), 57; Acta Pharm Sin B 2021, 11(5), 1341-1354; Adv Healthc Mater 2021, 10(20), e2100812; J Control Release 2021, 329, 934-947; Acta Pharm Sin B 2021, 11(12), 4032-4044; Mol Pharm 2021, 18(7), 2694-2702.

Author Response

Authors response: Compiled. The authors thank valued reviewer#1 for their valuable suggestions and comments, which certainly helped us in improving the quality of our manuscript. The authors also thank the reviewer for regarding the manuscript to be covering a important topic in brain delivery. As per the recommendations and suggestions of the reviewer, we have revised the manuscript by prudently following all the below-mentioned comments. A detailed point-by-point response to the reviewer’s suggestions/ comments are attached with the revised manuscript.

Reviewer 2 Report

This is an extensive and multifaceted work, really broadly covering the topic. The fact that the authors tried to characterize all currently known methods of drug delivery through the BBB is both a strength and a weakness of the work at the same time. Strong, as the many delivery technologies discussed allow the reader to choose a method according to their interests and then follow the links to the publications. Weak, as different sections are written at different levels of quality.

In general, I recommend this manuscript for publication after the authors consider the comments below:

1. Figures 1 and 2, in my opinion, even taking into account their schematic nature, do not correctly reflect the structure of the BBB. From Figure 1, one can conclude that astrocytes (entirely) located inside the BBB, while only their endfeet are part of it. Neurons are depicted  located away the astrocytes, while they are intertwined with astrocytes to form neurovascular units. In Figure 2, the pericyte is shaped like an astrocyte because it has a endfoot, while the astrocyte does not. The crosssection of the vessel corresponds to the arteriole, since it has a layer of smooth muscle cells and therefore does not relevant to the BBB, which surrounds the capillaries. All of the above makes Figures 1 and 2 misleading and makes them mistrustful.

2. In section 3.1, it is not clear how the phrase "Compressive, tensile, and shear stress are mechanical stress that can be generated internally or applied externally with relative ease" relates to drug delivery across the BBB. At the same time, among the mechanical ways to overcome the BBB, the authors did not mention works on the use of loud sound and music for this purpose. Authors can  just try Google Schoolar search for "BBB sound" or "BBB music".

3. End of page 8: Repeated  sentence: "BBB rapidly and directly within only 30 minutes of injection. Specific cytokines can cross the BBB rapidly and directly within only 30 minutes of injection".

4. What did the authors mean by the phrase " thermodynamics play a role in controlling transcytosis " ? 

5. Page 40, top: "The brain's lymphatic system sometimes referred to as the glymphatic system". This is an incorrect statement. There are meningeal lymphatic vessels, the presence of which is proven and recognized, while the "glymphatic system" is a hypothesis of a mechanism for clearing harmful metabolites from the parenchyma, in some essential parts not proven and not recognized by many researchers. Therefore, the content of section 7 does not reflect the current state of the topic. I encourage authors to check out one of the recent balanced reviews on the subject, such as doi.org/10.1186/s12987-021-00282-z.

Author Response

Authors response: Compiled. Authors thank valued reviewer#2 their valuable suggestions and comments, which certainly helped us in improving the quality of our manuscript. The reviewer’s recommendation to publish our manuscript is highly encouraging to our team. As per the recommendations and suggestions of reviewer, we have revised the manuscript by prudently following all the below mentioned comments. A detailed point-by-point response to the reviewer’s suggestions/ comments are submitted with the revised manuscript.

Reviewer 3 Report

I advise these fine authors to add a consideration that the LIVER is the major issue limiting drug delivery  for many pain drugs due to First Pass Effect.

Also, most authors forget , as these fine authors did, that the BBB is a series of endothelial cells that have mg levels of membrane associated and soluble proteases  associated with these cells. These enzymes  severely limit the delivery of any drug or molecule that is a peptide or even a small protein. They are proteolytically metabolized at the endothelial cells (BBB) before they pass to brain. This is the reason many Big Pharma protein drugs are pegylated.

Author Response

Authors response: Compiled. The authors thank the valued reviewer#3 for their valuable suggestions and comments, which certainly helped us in improving the quality of our manuscript. Authors thank the expert reviewer #3 for citing key igniting topics concerning brain delivery of drugs. Authors agree that the BBB also comprises membrane-associated soluble proteases, a prime factor that severely limits the delivery of therapeutics. It is also responsible for catalyzing therapeutics' metabolism before they actually cross the brain. This factor must also be considered while designing a drug discovery program for the brain delivery of therapeutics. Agreeing with the suggestion, the authors have incorporated this discussion point concerning the membrane-associated soluble proteases on the BBB endothelial cells.